# Single-Atom Ce-N-C Nanozyme Ameliorates Type 2 Diabetes Mellitus by Improving Glucose Metabolism Disorders and Reducing Oxidative Stress

**DOI:** 10.3390/biom14091193

**Published:** 2024-09-22

**Authors:** Yitong Lin, Yanan Wang, Qi Zhang, Ruxin Gao, Fei Chang, Boran Li, Kunlun Huang, Nan Cheng, Xiaoyun He

**Affiliations:** 1Key Laboratory of Precision Nutrition and Food Quality, Ministry of Education, College of Food Science and Nutritional Engineering, China Agricultural University, Beijing 100083, China; linyitong1999@163.com (Y.L.); wangyanan_cau@126.com (Y.W.); zhangqi20182154@163.com (Q.Z.); ruxin1958@163.com (R.G.); ch_angfei@163.com (F.C.); cl888246@163.com (B.L.); foodsafety66@cau.edu.cn (K.H.); 2Key Laboratory of Functional Dairy, Ministry of Education, College of Food Science and Nutritional Engineering, China Agricultural University, Beijing 100083, China; 3Key Laboratory of Safety Assessment of Genetically Modified Organism (Food Safety), The Ministry of Agriculture and Rural Affairs of China, Beijing 100083, China

**Keywords:** single-atom nanozyme, type 2 diabetes mellitus, glycogen synthesis, insulin resistance, oxidative stress

## Abstract

Type 2 diabetes mellitus (T2DM) as a chronic metabolic disease has become a global public health problem. Insulin resistance (IR) is the main pathogenesis of T2DM. Oxidative stress refers to an imbalance between free radical production and the antioxidant system, causing insulin resistance and contributing to the development of T2DM via several molecular mechanisms. Besides, the reduction in hepatic glycogen synthesis also leads to a decrease in peripheral insulin sensitivity. Thus, reducing oxidative stress and promoting glycogen synthesis are both targets for improving insulin resistance and treating T2DM. The current study aims to investigate the pharmacological effects of single-atom Ce-N-C nanozyme (SACe-N-C) on the improvement of insulin resistance and to elucidate its underlying mechanisms using HFD/STZ-induced C57BL/6J mice and insulin-resistant HepG2 cells. The results indicate that SACe-N-C significantly improves hepatic glycogen synthesis and reduces oxidative stress, as well as pancreatic and liver injury. Specifically, compared to the T2DM model group, fasting blood glucose decreased by 29%, hepatic glycogen synthesis increased by 17.13%, and insulin secretion increased by 18.87%. The sod and GPx in the liver increased by 17.80% and 25.28%, respectively. In terms of mechanism, SACe-N-C modulated glycogen synthesis through the PI3K/AKT/GSK3β signaling pathway and activated the Keap1/Nrf2 pathway to alleviate oxidative stress. Collectively, this study suggests that SACe-N-C has the potential to treat T2DM.

## 1. Introduction

According to the latest statistics from the International Diabetes Federation (IDF), the global diabetes prevalence will climb to 783 million people in 2045 compared to the estimated data of 537 million individuals in 2021. The incidence of diabetes mellitus is rapidly increasing, and diabetes mellitus has become a global health concern [1]. Diabetes mellitus is a complex chronic metabolic disease mainly categorized into four types: type 1 diabetes mellitus, type 2 diabetes mellitus (T2DM), gestational diabetes mellitus and other specific types of diabetes mellitus [2]. Among them, T2DM is the most common type of diabetes, accounting for more than 90% of all patients with diabetes [3]. T2DM is usually accompanied by multiple acute and chronic complications [4], which seriously affect human health and have become a major challenge in contemporary medicine. Therefore, finding effective treatment methods and exploring its pathogenesis has recently become a research hotspot at home and abroad.

T2DM, characterized by hyperglycemia, is triggered by IR and deficiency in insulin secretion. The major driving factors of the global T2DM epidemic include obesity, a sedentary lifestyle, increased consumption of unhealthy diets [5], genetics and the environment [6]. Affected by various factors, the sensitivity of the target organs (liver, muscle and adipose tissue) to insulin decreases, which leads to IR. At this time, the islet cells can still secrete insulin compensatively, but it causes hyperinsulinemia. As IR worsens, pancreatic function is impaired and insulin secretion decreases [7]. Different studies suggest that the mechanism of inducing T2DM is very complex and diverse, and the pathogenesis of T2DM has not been fully understood yet. However, many studies have shown that IR is one of the key triggers of T2DM [4,8,9].

The occurrence of IR is mainly associated with the blockade of insulin signaling, the dysregulation of glycogen synthesis and the appearance of oxidative stress, which subsequently lead to a decrease in glucose transport [10,11,12]. The liver, as the first organ that insulin reaches after being secreted in the pancreas, is essential for maintaining normal glucose homeostasis [13]. The liver mediates insulin primarily through the PI3K/AKT pathway. AKT is a key regulator in the insulin signaling pathway which can regulate glucose metabolism through multiple pathways. AKT binds to its substrate, AS160, to promote the membrane transport of glucose transporter type 4 (GLUT4) and accelerate glucose consumption [14]. AKT inhibits the expression of forkhead box O (FoxO) by phosphorylating three conserved residues of FoxO, thereby reducing gluconeogenesis and enhancing insulin signaling [15]. The stimulation of AKT phosphorylates glycogen synthase kinase 3β (GSK3β), which further activates glycogen synthetase (GS), thereby promoting glycogen synthesis [16]. Evidence has indicated that oxidative stress decreases insulin sensitivity in peripheral tissues [17] and induces IR through multiple molecular mechanisms, promoting the development of T2DM. Rehman et al. [18] proposed that oxidative stress is responsible for insulin resistance, impaired insulin secretion and glucose utilization, abnormal hepatic glucose production and, ultimately, overt T2DM. In addition, the Keap1/Nrf2 pathway is the principal protective response to oxidative stress [19] and represents an attractive therapeutic target in hepatic injury in T2DM [20]. Therefore, T2DM can be alleviated by modulating glycogen synthesis as well as oxidative stress.

Due to their excellent catalytic activity and specificity, enzymes have been widely used for different purposes, but they are costly and unstable. With the progress of nanotechnology, more and more enzyme mimics based on nanomaterials have been discovered. Scrimin et al. [21] put forward the term “nanozyme” in 2004. Since then, a large number of nanozymes have been developed, such as metal and metal oxide nanomaterials, carbon nanomaterials and metal-organic frameworks [22]. However, the utilization of metal active centers is relatively low, leading to unfavorable catalytic activity. With the advancement of synthesis technology, Zhang et al. [23] prepared an iron oxide-Pt1/FeOx catalyst loaded with only single Pt atoms and proposed the term “single-atom catalyst” for the first time. Single-atom nanozymes (SANs) are nanoparticle catalysts with enzyme-like activity [24], in which the metal active sites are uniformly dispersed in the material and have no obvious interaction with each other [25], which improves the atomic utilization efficiency and the density of the active site [26]. Due to these properties, SANs exhibit higher catalytic activity than conventional nanozymes [27]. Because of their enzyme-like activities, such as peroxidase, catalase, superoxide dismutase, glutathione peroxidase and oxidase, many SAzymes are used in the field of biomedicine, especially in areas such as antibacterial [28], antitumor [29] and anti-inflammatory [30] treatment. Little is known about the applications of SANs in treating metabolic diseases.

The object of this study is SACe-N-C with a novel metal active center. SACe-N-C was synthesized by surfactant template induction combined with high-temperature pyrolysis. The characterization results show that the SACe-N-C holds a wirelike structure with a diameter of ∼40 nm, and Ce are demonstrated to be present in SACe-N-C as an atomically dispersed form [31]. Different studies have shown that SACe-N-C can rapidly detect organophosphorus and carbamate pesticide residues [32] and can be used as oxygen reduction reaction electrocatalysts for batteries [31]. Besides, the SACe-N-C nanozyme had high oxygen reduction reaction activity and good stability. In our previous study, we found that it improved hyperglycemia in high-fat diet (HFD)-induced T2DM mice [33]; however, the ability to improve insulin resistance with relatively inadequate insulin secretion is not clear. In this study, a T2DM model induced by HFD and streptozotocin (STZ) in mice and an IR-HepG2 cell model were constructed, and the molecular mechanism of SACe-N-C in the protection against T2DM was clarified.

## 2. Materials and Methods

### 2.1. Materials

SACe-N-C was supplied by Associate Professor Li Jincheng from Kunming University of Science and Technology, and its preparation and characterization can be referred to in our previous study [31,33]. The high-fat diet (HFD) (60% energy) was purchased from Research Diets (D12492, New Brunswick, NJ, USA). The maintenance diet (13.5% energy) was obtained from Beijing Huafukang Co., Ltd. (H10060, Beijing, China). Streptozotocin (STZ) was purchased from Sigma-Aldrich (S0130, Saint. Louis, MO, USA).

### 2.2. Animal Experiments

The animal program was approved by the Animal Ethics Committee of China Agricultural University (approval number: AW22203202-4-10). The six-week-old male C57BL/6J mice weighing 20.0 ± 2.0 g were purchased from Beijing Charles River Laboratories (Beijing, China) and housed in an SPF animal house (22 ± 2 °C, 12 h light/dark cycles). The animal experiment design is shown in Figure 1A. After one week of acclimatization, 10 mice were randomly selected as the control (Chow) group (n = 10) and were fed the maintenance diet. Other mice were fed the HFD for six weeks. Then, the T2DM model was established by intraperitoneal injection (i.p.) of low-dose STZ (90 mg/kg B.W.). One week later, the fasting blood glucose (FBG) level of all mice was measured by tail tip. Mice with an FBG level of 11.1 mmol/L or above were considered successful models and were randomly divided into two groups: the model group (n = 10) and the SACe-N-C group (n = 10). The Chow and model group received i.p. of saline, and the SACe-N-C group received i.p. of SACe-N-C solution (10 mg/kg) daily. The body weight and the FBG of each mouse were monitored weekly. After five weeks, all animals were fasted for 6 h (8:00 a.m.–2:00 p.m.) and then sacrificed. Their organs were dissected immediately.

### 2.3. Glucose- and Insulin-Tolerance Tests and Serum Insulin Content Determination

The mice were fasted for 6 h (8:00 a.m.–2:00 p.m.), and then, they were administered glucose orally (2 g/kg) [34]. ITT were determined following i.p. injection of insulin (0.75 U/kg). Blood glucose levels were determined in samples taken from the tail vein at 15, 30, 60, 90 and 120 min using a glucometer. Serum insulin levels were measured using a mouse insulin ELISA kit (CSB-E05071m, CUSBIO, Wuhan, China) [35].

### 2.4. Measurement of Biochemical Indications in Mice

At the end of the experiment, blood taken from the orbits was centrifuged at 3500 rpm for 15 min, and the supernatant was taken as serum. A biochemistry analyzer was used to determine the levels of triglyceride (TG), cholesterol (TC), UREA, aspartate aminotransferase (AST), alanine aminotransferase (ALT), low-density lipoprotein (LDL), high-density lipoprotein (HDL), Glucose (Glu) and lactic dehydrogenase (LDH).

### 2.5. Histological Analysis

The fixed liver and pancreas were dehydrated, embedded, sectioned and then stained with hematoxylin and eosin (H&E). Then, sections were observed under a microscope (DM2500, Leica, Wetzlar, Germany).

### 2.6. Establishment of Insulin Resistance Cell Model

HepG2 cells (Cell Resource Center, Peking Union Medical College (PCRC)) were cultured in Dulbecco’s Modified Eagle Medium (DMEM) supplemented with 10% fetal bovine serum at 37 °C and 5% CO_2_ conditions. The cells were seeded in 96-well plates at a density of 10^4^ cells/well, and after 50–60% confluence, the cells were cultured with medium containing different concentrations of palmitic acid (PA) (0, 200, 250, 300, 400, 500, 600, 800 µM). After 24 h, the glucose content in the supernatant was determined by the glucose oxidase method.

### 2.7. Measurement of Cell Viability

HepG2 cells were seeded into 96-well plates and treated with different concentrations of PA (0, 200, 250, 300, 400, 500, 600, 800 µM) and SACe-N-C (0, 3.125, 6.25, 12.5, 25, 50 μg/mL) for 24 h, and cell viability was detected according to Cell Counting Kit-8 (C0038, Beyotime, Shanghai, China).

### 2.8. Measurement of Glycogen Content and Glucose Uptake

The cells were seeded in 6-well plates, and after 50–60% confluence, the cells were treated with 300 μM PA and 12.5 μg/mL SACe-N-C for 24 h. The effect of SACe-N-C on glycogen synthesis ability was determined by a glycogen assay kit (A043-1-1, Nanjing Jiancheng Bioengineering Institute, Nanjing, China).

The effect of SACe-N-C on glucose uptake in IR-HepG2 cells was determined by fluorescence glucose 2-NBDG. After treatment, the cells were washed three times with PBS. Sugar-free DMEM medium containing 200 μM 2-NBDG was added to each well, and they were incubated for 30 min. Then, each well was washed three times with PBS to stop glucose uptake and to ensure that no 2-NDBG remained [36]. The fluorescence intensity of 2-NBDG was measured by flow cytometry (BD FACSalibur, Bergen County, NJ, USA).

### 2.9. Measurement of Oxidative Stress Index

Malonaldehyde (MDA), Super Oxide Dismutase (SOD) and Glutathione peroxidase (GPx) activities of serum and liver homogenate were detected according to the instructions of the kit.

### 2.10. Quantitative Real-Time Polymerase Chain Reaction (qRT-PCR)

Total RNA was extracted from liver tissues or HepG2 cells using Trizol (15596018, Ambion, Austin, TX, USA) and quantified using a NanoDrop™ spectrophotometer (Thermo Scientific, Waltham, MA, USA), and cDNA was synthesized using a kit (AT311, TransGen Biotech, Beijing, China). The mRNA levels of the target genes were then detected using the FastKing One-Step RT-PCR Kit. (FP 205, TIANGEN Biotech, Beijing, China). Primer sequences are listed in Table 1.

### 2.11. Statistical Analysis

The results are presented as the mean ± standard error of the mean (SEM). All data differences were analyzed by one-way ANOVA (Tukey’s multiple comparison tests) on GraphPad Prism 8.0, which was considered significant at *p* < 0.05.

## 3. Results

### 3.1. Effect of SACe-N-C on Body Weight, Food Intake and Fasting Blood Glucose (FBG) in HFD/STZ-Induced T2DM Mice

As shown in Figure 1A, we constructed a T2DM mouse model by HFD combined with low-dose STZ (90 mg/kg). The mice gained weight in the first six weeks, which was significantly higher than the Chow group in the third week. After injection of STZ, mice showed a decrease in body weight, suggesting that STZ may have produced an acute injury in mice. The SACe-N-C intervention began in week eight, and there were no significant effects (*p* > 0.05) on body weight (Figure 1B) and food intake (Figure 1C) during the treatment period.

Our results show that the level of FBG was remarkably elevated in the diabetic mice. After administration, the FBG levels in the SACe-N-C group decreased to 9.97 ± 1.22 mmol/L, which was significantly lower than that in the model group. (Figure 1D). In addition, SACe-N-C treatment reduced the glucose concentration in the serum of T2DM mice (Figure 1E).

### 3.2. SACe-N-C Improved Glucose Tolerance, Insulin Sensitivity and Pancreatic Injury in HFD/STZ-Induced T2DM Mice

As shown in Figure 2A, after 15 min of glucose stimulation, the blood glucose level of T2DM mice increased significantly compared with the control group, indicating an impaired glucose tolerance in the diabetic mice, while treatment with SACe-N-C could decrease the glucose excursions. Besides, the area under the curve (AUC) decreased by 23.28% in the SACe-N-C group relative to the model group (*p* < 0.05) (Figure 2B), indicating that SACe-N-C may improve the glucose tolerance of T2DM mice. After insulin injection, the blood glucose level in the model group was higher than in other groups at all time points, indicating that the T2DM mice showed symptoms of insulin resistance. After SACe-N-C treatment, blood glucose level and AUC were both lower than the model group (Figure 2C,D). SACe-N-C improved the insulin resistance of T2DM mice. To reduce the effect of initial differences in blood glucose, we normalized fasting blood glucose at 0 min, and the results more accurately indicate that the T2DM model was successfully established and that SACe-N-C had hypoglycemic effects (Appendix A).

Due to long-term insulin resistance, the function of islet cells in T2DM patients is gradually impaired, resulting in relatively insufficient insulin secretion, which has been proved by many studies [37,38,39]. The results of our experiment are similar. In contrast to the Chow group, the serum insulin level was significantly reduced in model mice, but was increased again after SACe-N-C treatment (Figure 2E). The H&E staining shows that the pancreatic cells of the Chow group were round or oval, with clear boundaries, uniform size and complete structure. In the T2DM mice, STZ induced compensatory hypertrophy of pancreatic cells, and islets showed irregular contours. However, these phenomena were obviously reversed after SACe-N-C treatment; the injury of the pancreas was improved, the morphology of cells became clear and the cells were arranged compactly (Figure 2F). These findings suggest that SACe-N-C improves glucose tolerance, attenuates IR, alleviates islet functional impairment, and promotes insulin secretion in T2DM mice.

### 3.3. SACe-N-C Improved Liver Injury in HFD/STZ-Induced T2DM Mice

In our experiment, the liver index of mice in the model group exhibited a tendency to increase compared with that in the Chow group, although there was no significant difference, which may be associated with weight gain. However, following treatment with SACe-N-C, the liver index decreased significantly (Figure 3A). Compared with the Chow group, the serum ALT and LDH in the model group were significantly increased and AST showed an upward trend, indicating liver injury in T2DM mice. However, SACe-N-C could significantly reverse this phenomenon, as ALT and LDH levels were significantly decreased (*p* < 0.01) and the AST level approached that of the control group. (Figure 3B).

Typically, a portion of blood glucose is converted into glycogen for storage, which is one method to reduce blood sugar levels. Figure 3C illustrates that SACe-N-C increases hepatic glycogen synthesis.

To investigate the effects of SACe-N-C on lipid metabolism in T2DM mice, the blood lipid index was determined. The results indicate that SACe-N-C treatment significantly reversed the increase in serum TC and LDL concentrations in T2DM mice (*p* < 0.05) while having no significant effect on TG and HDL. This demonstrates that SACe-N-C can improve dyslipidemia in T2DM mice to some extent (Figure 3D). The H&E staining shows clear and intact hepatic lobules in the Chow group, with no steatosis. A large number of hepatocytes in the model group exhibited vacuolar-like steatosis. After SACe-N-C intervention, the number and area of fat vacuoles in liver tissue were reduced. In summary, SACe-N-C could effectively reverse HFD-induced liver injury (Figure 3E).

### 3.4. SACe-N-C Improved Oxidative Stress in HFD/STZ-Induced T2DM Mice

To evaluate the effect of SACe-N-C on the antioxidant capacity of T2DM mice, we first measured the levels of GPx, SOD and MDA in serum and liver tissue. As shown in Figure 4A–C, the level of SOD and GPx decreased significantly in the serum of the model group compared with the Chow group (*p* < 0.01), indicating that oxidative damage has occurred in T2DM mice. However, after five weeks of treatment, SACe-N-C ameliorated this reduction. Additionally, serum MDA level was significantly elevated in the model group, and treatment with SACe-N-C could reverse this change (*p* < 0.05). The changes in SOD, MDA and GPx in the liver were similar (Figure 4D–F).

To explore the potential antioxidant mechanism of SACe-N-C, we measured the relative mRNA expression of relevant genes in the liver by qRT-PCR. The results showed that mRNA levels of *Nrf2* and *Keap1* in the model group were significantly decreased, while they were increased in the SACe-N-C group. Besides, antioxidant genes *HO-1* and *SOD1* mRNA levels, which were suppressed by HFD/STZ, were effectively reversed by SACe-N-C intervention (Figure 4G). These results indicate that SACe-N-C treatment can ameliorate oxidative stress through the Keap1/Nrf2 pathway in vivo.

### 3.5. Effect of SACe-N-C on Insulin Signaling Pathway In Vivo

Under normal physiological conditions, insulin released after meals activates the PI3K/AKT signaling pathway, enhances glucose utilization, decreases gluconeogenesis and maintains glucose metabolic balance [40]. Inhibition of the PI3K/AKT signaling pathway leads to IR and T2DM. We first used qRT-PCR to quantify the expression levels of related genes in the livers. Interestingly, treatment with SACe-N-C significantly increased the expression of the key genes, including *IRS-1*, *PI3K*, *AKT* and *GSK3β* (Figure 5).

### 3.6. Establishment of IR-HepG2 Cell Model and Effect of SACe-N-C on Glucose Uptake and Consumption

Next, we examined the effect of SACe-N-C on the cellular insulin resistance model. In this study, HepG2 cells were treated with different concentrations of PA and cell viability was assessed after 24 h. The concentrations of 200, 250 and 300 μmol/L PA showed no toxic effect on HepG2 cells (Figure 6A). To establish the IR-HepG2 cell model, we determined the effect of three safe concentrations of PA on glucose consumption. Our findings indicated that treatment with 300 μmol/L PA led to the lowest glucose consumption (Figure 6B) and inhibited the expression of PI3K and AKT (Figure 6C). Consequently, 300 μmol/L PA was selected to induce the IR-HepG2 model. Subsequently, we determined the optimal concentration of SACe-N-C to be 12.5 μg/mL by CCK-8 assay (Figure 6D,E).

To investigate the effect of SACe-N-C on glucose uptake in IR-HepG2 cells, we initially measured the glucose content in the cell supernatant. The results show that the glucose consumption of IR-HepG2 cells was decreased compared with untreated cells (*p* < 0.05). However, this phenomenon was reversed following treatment with SACe-N-C (*p* < 0.05) (Figure 6F). To further validate the glucose consumption effect of SACe-N-C, the glucose uptake experiment was performed. A 2-NBDG assay was employed to measure glucose uptake. The 2-NBDG uptake was markedly lower in IR-HepG2 cells (Figure 6G,H). However, SACe-N-C treatment effectively reversed this activity. Inhibition of glycogen synthesis is another distinctive feature of IR [41]. Our results demonstrate that PA induced a rapid decrease in glycogen content in IR-HepG2 cells, while SACe-N-C treatment elevated glycogen content (Figure 6I).

### 3.7. Effect of SACe-N-C on Oxidative Stress and Insulin Signaling Pathway In Vitro

Several studies have shown that PA can induce oxidative stress in vitro [42,43,44]. To evaluate the protective effect of SACe-N-C against oxidative stress, HepG2 cells were cultured with and without 300 mM of PA and SACe-N-C (12.5 μg/mL) for 24 h. Subsequently, MDA level and SOD and GPx activity were evaluated. Our findings suggest that a significant increase in MDA level and a remarkable decrease in SOD and GPx activity was observed in PA-exposed HepG2 cells. However, SACe-N-C reversed these results. (Figure 7A–C).

As shown in Figure 7D, the mRNA levels of *Keap1* and *Nrf2* were decreased within 24 h following exposure to 300 mM of PA. Furthermore, the decreased mRNA levels of *HO-1*, *NQO1*, *GPx1* and *SOD1* were also observed in PA-incubated HepG2 cells. This situation was reversed by SACe-N-C. Different studies have shown that substances with antioxidant properties can improve oxidative stress and thereby alleviate T2DM [45]. In our previous study, it was proved that SACe-N-C exhibits antioxidant enzyme activity. We observed that after the intervention with SACe-N-C, the expression of Nrf2 and antioxidant enzyme genes in liver tissues and IR-HepG2 cells were significantly up-regulated compared with the model group, indicating that SACe-N-C alleviates oxidative stress by activating the Keap1/Nrf2 pathway in the liver tissue.

The effects of SACe-N-C on the insulin signaling pathway in IR-HepG2 cells were evaluated by qRT-PCR. Similar to the results in vivo, the mRNA relative expression levels of *IRS1*, *PI3K*, *AKT* and *GSK3β* in IR-HepG2 cells were significantly decreased (*p* < 0.05). However, it was increased after treatment with SACe-N-C (Figure 7E). These results indicate that SACe-N-C can improve glucose metabolism via the PI3K/AKT/GSK3β pathway.

## 4. Discussion

With the improvement of economic levels, the incidence of metabolic syndrome is rapidly increasing worldwide, posing a serious threat to human health and imposing a significant economic burden [46]. Long-term use of medications for the treatment of T2DM has been shown to lead to serious toxic side effects and drug resistance. Therefore, it is essential to screen and explore novel therapeutic agents for preventing and treating T2DM.

T2DM patients exhibit various symptoms at different stages, all of which are accompanied by elevated blood glucose levels [47]. In recent years, some researchers have reported that many nanozymes can alleviate high blood sugar levels of T2DM. For example, Zhou et al. [48] found that the administration of Fe_3_O_4_ nanozymes can reduce blood glucose levels and improve glucose tolerance and insulin sensitivity in T2DM mice. Similar results were obtained in this study. At present, most of the studies on the intervention of nanozymes in T2DM focus on controlling blood glucose levels [32,48], and there has been little discussion about the effect of SANs on pancreatic function. However, this study found that SACe-N-C improved pancreatic injury. The insulin secretion function of islet cells gradually declines for the duration of T2DM, decreasing at a rate of about 5% per year [49]. For T2DM patients, impaired islet function can be partially restored: for example, enhancing the sensitivity of the target organ to insulin can alleviate the secretion burden on islet cells, and any intervention that can reduce blood glucose level can enhance islet cell function [50]. Our findings suggest that both insulin secretion and islet cell damage improved following the SACe-N-C intervention, although the specific mechanism needs to be further explored.

In patients with obesity, type 2 diabetes is usually accompanied by fatty liver disease. It was found that the incidence of liver dysfunction in the general population is 8.74%, whereas in diabetes mellitus patients, it rises to 24.4% [51]. Generally, elevated ALT and AST levels are indicators of the degree of liver injury [52]. The T2DM mice established in this study also showed abnormal liver function, which was consistent with other relevant studies [53,54]. Furthermore, SACe-N-C can effectively lower ALT and AST levels in the serum of HFD/STZ-induced mice. The probability of abnormal blood lipid levels in T2DM patients is significantly higher than that in non-diabetic patients, and elevated blood lipid levels can lead to atherosclerosis, which may result in cardiovascular complications [55]. Additionally, TC, TG and LDL levels could indicate a change in liver lipid metabolism [56]. After treatment with SACe-N-C, the serum levels of TC and LDL were remarkably reduced. It demonstrates that SACe-N-C can regulate lipid metabolism in HFD/STZ-induced diabetic mice, which is consistent with our previous findings [33].

The liver plays an extremely crucial role in glucose metabolism [57]. Glucose is primarily cleared from the bloodstream through its conversion to glycogen [58]. From our data, it is evident that IR causes impaired absorption and transformation of glucose, thereby reducing its uptake. Additionally, there is an increased glycogen content in hepatocytes following SACe-N-C treatment, both in vitro and in vivo. Krssak M et al. [59] found that T2DM patients exhibit a reduction in hepatic glycogen synthesis, which is consistent with our results. The PI3K/Akt/GSK3β pathway is an important insulin signaling pathway [60]. Evidence shows that AKT is a key node, which can affect the activity of GSK3β to promote glucose utilization by cells [61]. In this study, SACe-N-C treatment observably increased the expression of *PI3K*, *Akt* and *GSK3β*, indicating that SACe-N-C intervention alleviates glucose metabolism disorders by regulating the PI3K/AKT/GSK3β signaling pathway.

Oxidative stress is related to T2DM, leading to glucose intolerance, IR and β-cell dysfunction, etc. [62]. MDA is a product of lipid peroxidation which is often used to reflect the degree of oxidative damage and is an important indicator of oxidative stress [63]. The results of our study revealed a 36% and a 20% reduction in MDA levels in the liver and IR-HepG2 cells following treatment with SACe-N-C, respectively. SOD and GPx are crucial antioxidant enzymes that remove excessive free radicals from the body [64]. In this study, we found that the GPx activity in the liver of HFD/STZ-induced T2DM mice was elevated by the SACe-N-C treatment, while SACe-N-C also enhanced the SOD activity in serum. In addition, the Keap1/Nrf2 system plays an important role in the prevention of T2DM [20]. Nrf2 is a key transcription factor in regulating oxidative stress [65]. The activation of Nrf2 can initiate the transcription of downstream antioxidant enzyme genes such as SOD, GPx, HO-1 and NQO-1, thereby improving oxidative damage [66]. Decreased Nrf2 levels increase blood glucose levels, worsen glucose intolerance and impair insulin signaling [20]. Similarly, we observed inhibition of Nrf2 expression both in vivo and in vitro. Different studies have shown that substances with antioxidant properties can improve oxidative stress and thus alleviate T2DM [45]. In our previous study, it was proved that SACe-N-C has antioxidant enzyme activity. After SACe-N-C intervention, we observed significant upregulation of Nrf2 and antioxidant enzyme genes in liver tissues and IR-HepG2 cells compared with the model group, indicating that SACe-N-C alleviates oxidative stress by activating the Keap1/Nrf2 pathway in the liver tissue.

Cerium-based nanozymes can specifically scavenge ROS and protect tissues from damage caused by excessive ROS [67]. T2DM is typically associated with increased ROS levels, and SACe-N-C can reduce intracellular ROS by exerting SOD and CAT enzyme activities. In addition, ROS at normal physiological levels can activate Nrf2 by activating the PI3K/AKT signaling pathway [68,69]. Therefore, this is the mechanism by which SACe-N-C activates the PI3K/AKT/GSK3β and Keap1/Nrf2 pathways.

SANs offer a great potential for diverse applications. However, to use SANs in the biomedical field, their biosafety evaluation is essential. In this study, the SACe-N-C group received daily intraperitoneal injections of 10 mg/kg SACe-N-C for 4 weeks. During this period, no significant difference was observed in the SACe-N-C treatment group compared with normal mice. This demonstrates that SACe-N-C has good biosafety. However, long-term biosafety still needs to be further evaluated.

In conclusion, our results show that SACe-N-C plays a critical role in improving glycogen synthesis and oxidative stress in both HFD/STZ-induced mice and IR-HepG2 cells. Specifically, SACe-N-C intervention improved glycogen synthesis by regulating the PI3K/AKT/GSK3β signaling pathway. Furthermore, SACe-N-C ameliorates oxidative stress by regulating the Keap1/Nrf2 signaling pathway and the related antioxidant genes. Our findings provide new initiatives for the prevention and treatment of T2DM.

## Figures and Tables

**Figure 1 biomolecules-14-01193-f001:**
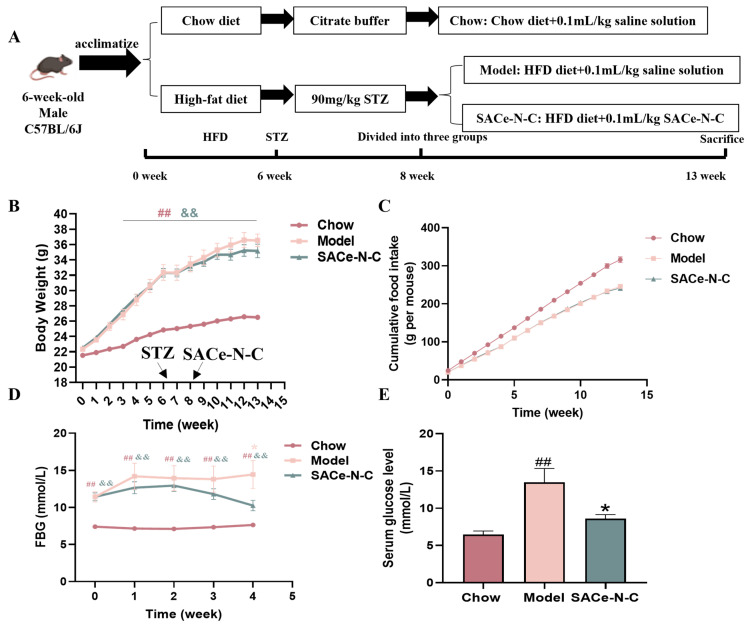
The effect of SACe-N-C on body weight, food intake and FBG in T2DM mice. (**A**) Experimental design. (**B**) Body weight gain curve. (**C**) Cumulative food intake. (**D**) FBG. (**E**) Serum glucose level. Data are expressed as mean ± SEM of n = 5–8 separate experiments. ^##^ *p* < 0.01 vs. Chow, * *p* < 0.05 vs. model, ^&&^ *p* < 0.05 vs. Chow.

**Figure 2 biomolecules-14-01193-f002:**
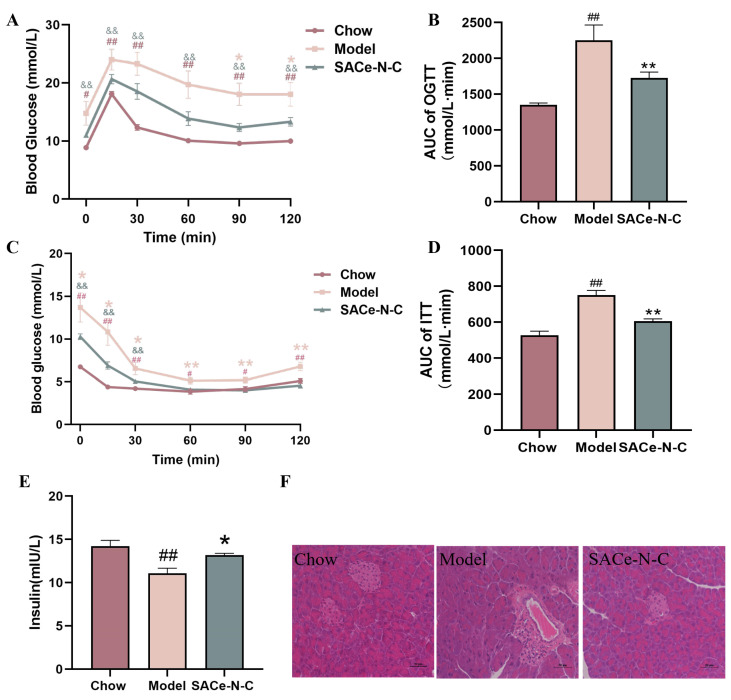
SACe-N-C improved glucose tolerance, insulin sensitivity and pancreatic injury in T2DM mice. (**A**) OGTT. (**B**) AUC of OGTT. (**C**) ITT. (**D**) AUC of ITT. (**E**) Serum insulin level. (**F**) H&E staining of the pancreas. The scale bar of H&E staining is 50 µm. Data are expressed as mean ± SEM of n = 5–8 separate experiments. ^#^
*p* < 0.05, ^##^
*p* < 0.01 vs. Chow, * *p* < 0.05, ** *p* < 0.01 vs. model, ^&&^ *p* < 0.05 vs. Chow.

**Figure 3 biomolecules-14-01193-f003:**
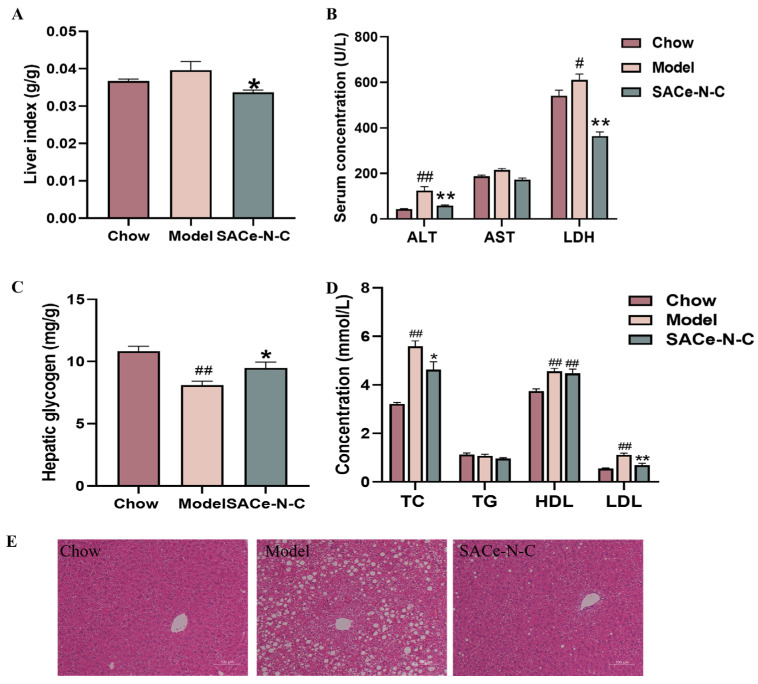
SACe-N-C improved liver injury in T2DM mice. (**A**) Liver index. (**B**) The serum levels of ALT, AST and LDH. (**C**) Glycogen content in the liver. (**D**) The Serum levels of TC, TG, HDL, LDL. (**E**) H&E staining of the liver. The scale bar of H&E staining is 100 µm. Data are expressed as mean ± SEM of n = 5–8 separate experiments. ^#^ *p* < 0.05, ^##^ *p* < 0.01 vs. Chow, * *p* < 0.05, ** *p* < 0.01 vs. model.

**Figure 4 biomolecules-14-01193-f004:**
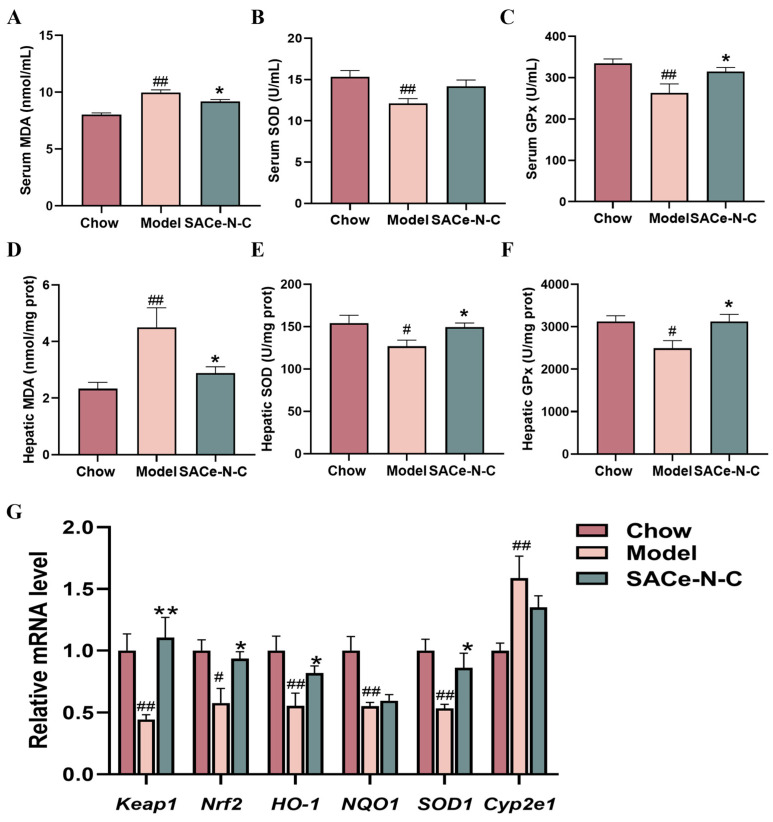
SACe-N-C improved oxidative stress in T2DM mice. (**A**–**C**) Indicators of oxidative stress in serum. (**D**–**F**) Indicators of oxidative stress in the liver. (**E**) qRT-PCR analysis of oxidative stress-related genes. (**G**) Relative mRNA levels of antioxidant genes. Data are expressed as mean ± SEM of n = 5–8 separate experiments. ^#^ *p* < 0.05, ^##^ *p* < 0.01 vs. Chow, * *p* < 0.05, ** *p* < 0.01 vs. model.

**Figure 5 biomolecules-14-01193-f005:**
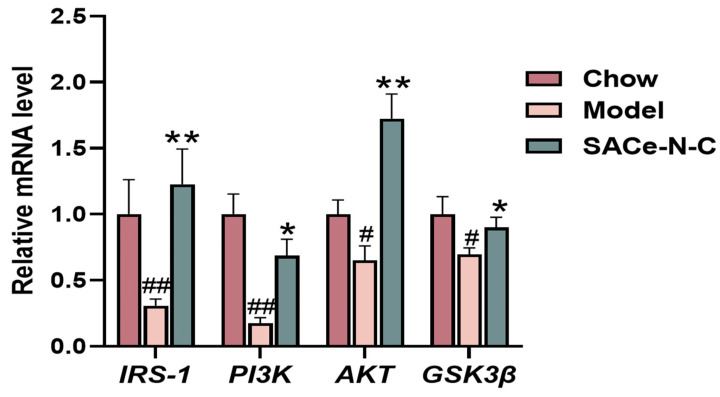
The relative mRNA expression of *IRS-1*, *PI3K*, *AKT* and *GSK3β* in the liver (n = 5–8). Data are expressed as mean ± SEM. ^#^ *p* < 0.05, ^##^ *p* < 0.01 vs. Chow, * *p* < 0.05, ** *p* < 0.01 vs. model.

**Figure 6 biomolecules-14-01193-f006:**
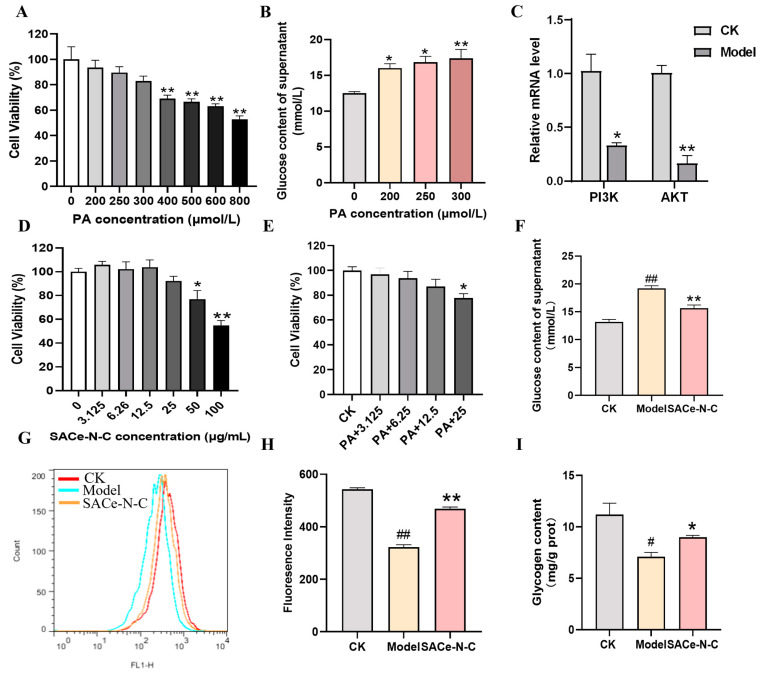
Establishment of IR-HepG2 cell model and effect of SACe-N-C on glucose uptake and consumption. Effects of different concentrations of PA on (**A**) HepG2 cell viability and (**B**) supernatant glucose consumption. (**C**) Effects of PA on mRNA levels of PI3K and AKT genes. (**D**,**E**) Effects of different concentrations of SACe-N-C with or without 300 μM PA on HepG2 cell viability. (**F**) Glucose content of cell supernatant. (**G**) The fluorescence intensity of 2-NBDG was measured by flow cytometer. (**H**) Quantification of fluorescence intensity. (**I**) Glycogen content. Data are expressed as mean ± SEM of n = 3 separate experiments. ^#^ *p* < 0.05, ^##^ *p* < 0.01 vs. CK, * *p* < 0.05, ** *p* < 0.01 vs. model.

**Figure 7 biomolecules-14-01193-f007:**
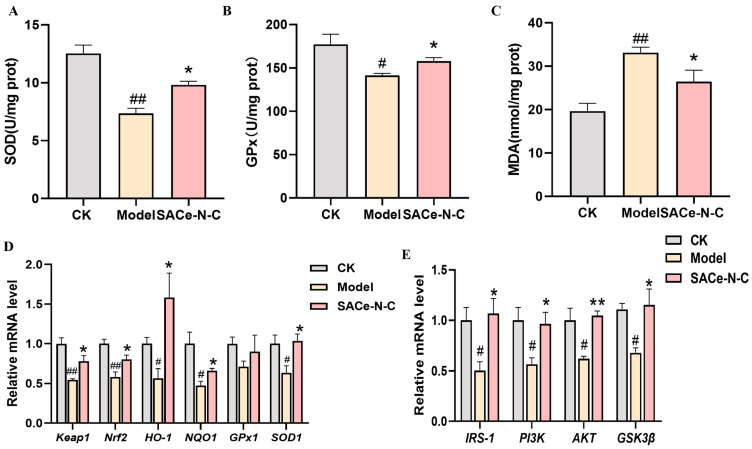
Effect of SACe-N-C on Oxidative Stress and Insulin Signaling Pathway in Vitro. (**A**,**B**) SOD and GPx activity in HepG2 cells incubated with or without PA, SACe-N-C. (**C**) MDA level in HepG2 cells incubated with or without PA, SACe-N-C. (**D**) The relative mRNA expression of oxidative stress-related genes in HepG2 cells. (**E**) The relative mRNA expression of *IRS-1*, *PI3K*, *AKT* and *GSK3β* in HepG2 cells. Data are expressed as mean ± SEM of n = 3 separate experiments. ^#^ *p* < 0.05, ^##^ *p* < 0.01 vs. CK, * *p* < 0.05, ** *p* < 0.01 vs. model.

**Table 1 biomolecules-14-01193-t001:** qRT-PCR primer sequences.

Species	Gene	Forward Primer	Reverse Primer
Mouse	*Keap1*	TGCCCCTGTGGTCAAAGTG	GGTTCGGTTACCGTCCTGC
	*Nrf2*	TCTTGGAGTAAGTCGAGAAGTGT	GTTGAAACTGAGCGAAAAAGGC
	*HO-1*	AAGCCGAGAATGCTGAGTTCA	GCCGTGTAGATATGGTACAAGGA
	*NQO1*	AGGATGGGAGGTACTCGAATC	AGGCGTCCTTCCTTATATGCTA
	*SOD1*	AACCAGTTGTGTTGTCAGGAC	CCACCATGTTTCTTAGAGTGAGG
	*Cyp2e1*	CGTTGCCTTGCTTGTCTGGA	AAGAAAGGAATTGGGAAAGGTCC
	*IRS-1*	CGATGGCTTCTCAGACGTG	CAGCCCGCTTGTTGATGTTG
	*PI3K*	ACACCACGGTTTGGACTATGG	GGCTACAGTAGTGGGCTTGG
	*AKT*	ATGAACGACGTAGCCATTGTG	TTGTAGCCAATAAAGGTGCCAT
	*GSK3β*	TGGCAGCAAGGTAACCACAG	CGGTTCTTAAATCGCTTGTCCTG
	*β-actin*	GGCTGTATTCCCCTCCATCG	CCAGTTGGTAACAATGCCATGT
Human	*Keap1*	GGAAACAGAGACGTGGACTTTCGTA	TCCAGGAACGTGTGACCATCATA
	*Nrf2*	GGTTGCCCACATTCCCAAAC	GCAAGCGACTCATGGTCATC
	*HO-1*	CCAGGCAGAGAATGCTGAGTTC	AAGACTGGGCTCTCCTTGTTGC
	*NQO1*	TGGTTTGGAGTCCCTGCCAT	CACTGCCTTCTTACTCCGGAAGG
	*GPx1*	CAGTCGGTGTATGCCTTCTCG	GAGGGACGCCACATTCTCG
	*SOD1*	GAGACTTGGGCAATGTGACTG	TTACACCACAAGCCAAACGA
	*IRS-1*	ACTGGACATCACAGCAGAATGA	AGAACGTGCAGTTCAGTCAA
	*PI3K*	CCACGACCATCATCAGGTGAA	CCTCACGGAGGCATTCTAAAGT
	*AKT*	TCTATGGCGCTGGAGATTG	TCTTAATGTGCCCGTCCTTG
	*GSK3β*	AGGAGAACCCAATGTTTCGTAT	ATCCCCTGGAAATATTGGTTGT
	*β-actin*	AGCCATGTACGTAGCCATCC	CTCTCAGCTGTGGTGGTGAA

## Data Availability

Data are available upon request from the corresponding authors.

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
