# Peer review of "Single-Atom Ce-N-C Nanozyme Ameliorates Type 2 Diabetes Mellitus by Improving Glucose Metabolism Disorders and Reducing Oxidative Stress"

_biomolecules, 2024, doi:10.3390/biom14091193_

Round 1
Reviewer 1 Report
Comments and Suggestions for Authors
The study showed that SACe-N-C plays an important role in improving glycogen synthesis by regulating the PI3K/AKT/GSK3β signalling pathway and reducing oxidative stress by regulating the Keap1/Nrf2 signalling pathway and related antioxidant genes in both HFD/STZ-induced mice and IR-HepG2 cells.
Here are some of the points that need to be addressed before consideration of the manuscript for publication in Biomolecules.
Title:
1. I suggest the title “Single-atom Ce-N-C nanozyme ameliorates Type 2 Diabetes Mellitus by improving glucose metabolism disorders and reducing oxidative stress” for the manuscript.
2. See the authors list, it's either incomplete or place “and” before Xiaoyun He.
Abstract:
3. The abstract presents the conclusion of the study but it does not contain specific results. Add the important results in quantitative terms.
4. It is not very clear what is meant by both in the sentence “Thus, both are targets for improving insulin resistance and treating T2DM”.
5. The sentence in lines 21 and 22, can be rephrased, i.e. “The results indicated that SACe-N-C significantly improved hepatic glycogen synthesis and reduced oxidative stress, as well as pancreatic and liver injury”.
6. The sentence in lines 24 and 25, can be rephrased to “Collectively, this study suggests that SACe-N-C has potential to treat Type 2 Diabetes Mellitus (T2DM).”
Introduction:
7. The introduction section covers the importance of DM and the underlying molecular events which lead to DM. Still, since the main objective of the study is to find an alternative treatment for DM, the introduction should include the existing treatment regimes and their shortcomings that the SACe-N-C would probably cover.
8. Have the authors confirmed the cytotoxicity of the nanoparticles? If they have checked previously, they should mention it here. Have the authors checked the effect of administering SACe-N-C to normal control mice, without T2DM? What are the potential long-term effects of using these SACe-N-C for diabetes management?
9. lines 48 and 49, “Studies believe that the mechanism of inducing T2DM is very complex and diverse, the pathogenesis of T2DM has not been clarified” can be rephrased to “Studies suggest that the mechanism of inducing T2DM is very complex and diverse, and the pathogenesis of T2DM has not yet been fully understood/explored”.
10. “To date, little research has been conducted on the metabolic diseases of SANs.”. can be rephrased to “Little is known about the applications of SANs in treating metabolic diseases”.
11. Line 33, replace “have” with “has”.
12. Line 42, add “is” after hyperglycemia.
13. line 45, replace “organ” with “organs”
14. line 46, replace “decreased” with “decreases”
15. Line 87, change “molecule” to “molecular” and “protection of T2DM” to “protection against T2DM”.
16. Line 101, change “was” to “is”
Methods:
17. Why didn’t the authors include any standard anti-diabetic drug as a control in the experiments?
18. Methods don’t mention the internal control genes for expression analysis of stress-related genes.
19. Line 131, correct “37ºC”
20. Elaborate all the abbreviations when they appear first in the manuscript.
21. Mention the catalogue and manufacturer of the “mouse insulin ELISA kit”,
22. Lines 145 to 150, change to past tense.
23. Italicize the names/symbols of all genes.
Results:
24. Figure 3B LDH, SACe-N-C treatment lowers LDH than Chow even, explain why?
25. Line 255, genes
26. Line 270, replace “containing” with “including”, add “and” before GSK3β
27. Line 307, (μg/ml) to (μg/mL)
Discussion:
28. The study shows that SACe-N-C can relieve DM by improving glycogen synthesis through the PI3K/AKT/GSK3β signalling pathway and reducing oxidative stress through the Keap1/Nrf2 signalling pathway, could the authors propose a model and add to the discussion section of MS that at what level the nanozyme interact to regulate these pathways/expression of genes.
29. What are the potential challenges in commercializing these nanozymes for diabetes treatment?
Regards
Comments on the Quality of English Languagethere are several English language and typo mistakes. The MS needs to be carefully checked for any English and typo errors.
Reviewer 2 Report
Comments and Suggestions for Authors
This study by Lin et al uses applies a SAN to a model of T2D mice. The authors show that SAN reduces several physiological parameters associated with T2D and as well as alter standard signaling pathways associated with T2D. Overall, this is an interesting paper, especially due to the new and innovative nature of SAN application to disease treatment. However, some items require attention before this paper can be published.
Major comments:
1) The introduction requires an expanded section on SANs and the particular SAN used in this study. These are new and unfamiliar to most readers, so the authors should thoroughly introduce them.
2) In the glucose tolerance test and insulin tolerance test, the authors to not normalize the zero time point, which likely skews the data. Normalizing the zero time point is standard practice for analysis of these tests and is required in order for publication.
Minor comments:
1) English usage could be improved in many places in the manuscript.
2) The final author appears to be missing.
Comments on the Quality of English LanguageEnglish usage could be improved in many places in the manuscript. There are some unusual word choices and sentence structures that are similar to red flags associated with AI-generated text. Please verify that AI was not used to generate portions of this manuscript.
Author Response
请参阅附件

Reviewer 3 Report
Comments and Suggestions for Authors
The manuscript titled " Single-atom Ce-N-C nanozyme Ameliorates Type 2 Diabetes 2 Mellitus by Improving Glucose Metabolism Disorders and 3 Oxidative Stress" demonstrates the pharmacological effects of single atom Ce-N-C nanozyme (SACe-N-C) on the improvement of insulin resistance and elucidate the underlying mechanisms using HFD/STZ-induced C57BL/6J mice and insulin-resistant HepG2 cells. The manuscript could be accepted for publication after minor revision:
Comments:
1. On what basis the author selected 12.5 µg/mL as the effective concentration of the Ce-N-C nanozyme for the insulin sensitivity study?
2. How did the author validate the establishment of insulin resistance in the in vitro cell model using palmitic acid (PA) in HepG2 cells?
3. In the cell viability study, the author reported that the single-atom Ce-N-C nanozyme was found to be safe at a concentration of 12.5 µg/mL. However, in a previous study, the author’s group reported 3.125 µM or 3.125 µg/mL as the safe concentration for the same material. The current finding contradicts the previous one. Please refer to Supplementary Figure S5 in your previous report for verification. https://spj.science.org/doi/10.34133/research.0095.
4. In lines 306-307, the author wrote, “HepG2 cells were cultured with or without 300 mM of PA and SACe-N-C (μg/mL) for 24 hours, and subsequently, MDA levels and SOD and GPx activities were evaluated”. Please verify the concentrations of PA and SACe-N-C used in the oxidative stress analysis study.
Reviewer 4 Report
Comments and Suggestions for Authors
The manuscript presents the effect of cerium-containing nanoparticles on diabetes in mice. The methodology is clearly described and the experimental work has been carefully carried out. The results are interesting but the long term impact is difficult to predict.
Round 2
Reviewer 2 Report
Comments and Suggestions for Authors
The authors have adequately addressed each of my concerns. The manuscript is now acceptable for publication.
Comments on the Quality of English LanguageEnglish usage is OK. A minor editorial review should take place prior to publication.